# Dynamics of *Mycobacterium tuberculosis* Lineages in Oman, 2009 to 2018

**DOI:** 10.3390/pathogens11050541

**Published:** 2022-05-04

**Authors:** Sara Al-Mahrouqi, Reham Ahmed, Saleh Al-Azri, Salama Al-Hamidhi, Abdullah A. Balkhair, Amina Al-Jardani, Amira Al-Fahdi, Laila Al-Balushi, Samia Al-Zadjali, Chamila Adikaram, Asmhan Al-Marhoubi, Amal Gadalla, Hamza A. Babiker

**Affiliations:** 1Biochemistry Department, College of Medicine and Health Sciences, Sultan Qaboos University, Muscat 123, Oman; saramahruqi@gmail.com (S.A.-M.); rihoahmed93@gmail.com (R.A.); salama.2012@hotmail.com (S.A.-H.); amira.alfahdi@gmail.com (A.A.-F.); s124973@student.squ.edu.om (A.A.-M.); 2National Tuberculosis Reference Laboratory, Ministry of Health, Muscat 100, Oman; laila.albalushi@gmail.com (L.A.-B.); samalzadjali@hotmail.com (S.A.-Z.); chamilaadhikaram@yahoo.com (C.A.); 3Central Public Health Laboratories, Ministry of Health, Muscat 100, Oman; alazri@gmail.com (S.A.-A.); aksaljardani@gmail.com (A.A.-J.); 4Department of Medicine, College of Medicine and Health Sciences, Sultan Qaboos University, Muscat 123, Oman; abdullag@gmail.com; 5Division of Population Medicine, School of Medicine, College of Biomedical Sciences, Cardiff University, Cardiff CF14 4XN, UK; gadalla@gmail.com; 6Institute of Immunology and Infection Research, University of Edinburgh, Edinburgh EH93LT, UK

**Keywords:** *Mycobacterium tuberculosis*, TB incidence, spoligotypes, MIRU-VNTR, Oman

## Abstract

Study aim. Effective Tuberculosis (TB) control measures in Oman have reduced the annual incidence of tuberculosis cases by 92% between 1981 and 2016. However, the current incidence remains above the program control target of <1 TB case per 100,000 population. This has been partly attributed to a high influx of migrants from countries with high TB burdens. The present study aimed to elucidate *Mycobacterium tuberculosis* infection dynamics among nationals and foreigners over a period of 10 years. **Methods**. The study examined TB cases reported between 2009 and 2018 and examined the spatial heterogeneity of TB cases and the distribution of *M. tuberculosis* genotypes defined by spoligotypes and MIRU-VNTR among Omanis and foreigners. **Results**. A total of 484 *spoligoprofiles* were detected among the examined isolates (n = 1295). These include 943 (72.8%) clustered and 352 (27.2%) unique isolates. Diverse *M. tuberculosis* lineages exist in all provinces in Oman, with most lineages shared between Omanis and foreigners. The most frequent spoligotypes were found to belong to EAI (318, 30.9%), CAS (310, 30.1%), T (154, 14.9%), and Beijing (88, 8.5%) lineages. However, the frequencies of these lineages differed between Omanis and foreigners. Of the clustered strains, 192 MTB isolates were further analysed via MIRU-VNTR. Each isolate exhibited a unique MIRU-VNTR profile, indicative of absence of ongoing transmission. **Conclusions**. TB incidence exhibits spatial heterogeneity across Oman, with high levels of diversity of *M. tuberculosis* lineages among Omanis and foreigners and sub-lineages shared between the two groups. However, MIRU-VNTR analysis ruled out ongoing transmission.

## 1. Introduction

TB remains a major global health problem, among the top 10 leading causes of death worldwide. 

Reduced access to health facilities, as a result of the COVID-19 pandemic in 2020, has resulted in the slowing down of declines in TB incidence and increased TB mortality, with an estimate of 1.3 million among HIV-negative people compared to 1.2 million in 2019 [1]. These trends are expected to become much worse in 2021 and 2022 [1]. 

In 2020, the global extent of TB incidence varied widely, from more than 500 per 100,000 to less than 5 per 100,000 population [2]. The Gulf Cooperation Council (GCC) countries (United Arab Emirates, Saudi Arabia, Qatar, Oman, Kuwait, and Bahrain) are in the low-incidence category (<10 cases per 100,000 population) and are well placed to target TB elimination. However, there are many variations within the GCC region, with a large proportion of cases among foreigners. In 2018, TB cases notified per 100,000 population was 31 in Qatar, 23 in Kuwait, 11 in Bahrain, 10 in Saudi Arabia, 6 in Oman, and 1 in UAE [3]. This is consistent with the pattern reported in 2014, among GCC countries, with the highest TB burden in Qatar (37 per 100,000 population) and the lowest in UAE (2.7 per 100,000 population) [4]. The differences between TB incidence in 2014 and 2018 reflects limited progress and obscure challenges in relation to the prospect of TB elimination in the region.

In Oman, successful control efforts have reduced the burden of TB by 85% in the last 25 years [5,6]. During the initial phase of the national TB control program, 1981–1990, case notifications decreased by 11% per year from 91 to 37 cases per 100,000. A further reduction was seen between 2001 and 2010 (12 to 10 cases per 100,000 population) and 2011 and 2016 (11 to 7.8 cases per 100,000 population) [5]. Between 2010 and 2016, 2352 new TB cases were reported: 1409 (60%) of them were Omanis and 943 (40%) were foreigners Most infected foreigners were from the Indian subcontinent, India (721 (76%)), Bangladesh (103 (11%)), and Pakistan (25 (3%)). Since 2017, the TB notification rate has remained between 8.5 and 5.9 cases per 100,000 [2] and persisted above the national program control target by 2035 of <1 per 100,000 population [6]. 

The TB epidemiology in Oman mirrors that in other Gulf Cooperation Council (GCC) countries [4]. In part, this is due to the high similarity in demographic structure and the high proportion of foreigners from countries with high TB burdens [7]. Migrants constitute a vast proportion of GCC countries’ populations, reaching >80% in Qatar and UAE [4]. In 1997, the Gulf Health Council (GHC) endorsed the goal of TB elimination, aiming at reducing the incidence of new cases among nationals to <1 per 10^4^ by 2010. However, no country in the region has achieved the desired targets [2]. This indicates the need for additional novel efforts to achieve the elimination targets, such as control of the spread of drug-resistant *M. tuberculosis* lineages and assessment of the role of foreigners from endemic areas in dissemination of the disease. 

A previous preliminary analysis has revealed a predominance of *M. tuberculosis* lineages commonly found in the Indian sub-continent, with a high proportion of shared lineages between Omanis and foreigners [8]. The present study extends these earlier findings to assess the spatial and temporal distribution of major *M. tuberculosis* lineages over a period of 10 years, 2009–2018. In addition, we examined the extent of genetic relatedness of *M. tuberculosis* lineages among Omanis and foreigners and possible transmission between the two groups.

## 2. Material and Methods

### 2.1. Mycobacterium tuberculosis Isolates

TB data from 2009 to 2018 were obtained from the National Tuberculosis Reference Laboratory (NRL) at the Central Public Health Laboratories (CPHL), Ministry of Health, Oman. TB is a notifiable disease in Oman, hence all cases across the country are reported to the CPHL. The laboratory provides TB diagnosis using standard microbiological procedures for the identification of the *M. tuberculosis* complex and in vitro drug susceptibility test (DST) for the administration of first-line anti-TB drugs to the population of Oman (4.4 million (2.8 nationals and 1.6 foreigners)) [9].

The population data used in this study were obtained from the National Centre for Statistics and Information (NCSI) based on the 2020 population census [9]. The data presented are aggregated for all provinces. A total of 2557 unique *M. tuberculosis* isolates were initially considered for this study; however, 18 were excluded, one BCG and 17 without DST. The remaining 2539 isolates were obtained from TB diagnosed cases, Omanis (n = 1341), and foreigners (n = 1198), between 2009 and 2018 (Figure 1). 

Ethical approval for the study was granted by the Medical Research and Ethics Committee (MREC) of the College of Medicine and Health Sciences, Sultan Qaboos University, Oman, under the reference number SQU-EC/075/18.

### 2.2. Spoligotyping and MIRU-VNTR Typing

Successfully sub-cultured isolates (n = 1295) were subjected to heat-killing for DNA extraction and subsequent Spacer Oligonucleotide Typing (spoligotyping) analysis [10]. These samples included MTB isolates from different years, different provinces, and both Omanis and foreigners. Spoligotyping (the classical 43-spacer format) was performed as previously described [11]. DNA samples of the *M. tuberculosis* H37Rv and *M. bovis* BCG strains were included as positive controls. Molecular biology-grade water was used as a negative control. The spoligotypes were then recorded in 43-digit binary format and compared with those recorded in the SITVIT2 database (http://www.pasteur-guadeloupe.fr:8081/SITVIT2/, accessed on 10 March 2022) to identify the spoligotype International Type (SIT) and family [12]. 

Out of the 943 clustered MTB isolates, 192 belonging to major clusters were randomly selected for Mycobacterial Interspersed Repetitive Unit-Variable Number Tandem Repeat (MIRU-VNTR) analysis. These samples comprised 45 clusters out of the total 132 clusters detected by spoligotyping, with 53 (24.9%) belonging to the two major clusters (Beijing/SIT 1 and CAS1-Delhi/SIT26). MIRU-VNTR typing based in the standard 24 loci was performed as described by Supply et al. (2006) on 192 *M. tuberculosis* samples to determine genetic relationships among isolates that belonged to 41 spoligo-lineages (clusters). First, each locus was amplified individually, as previously described. Then, the PCR products were detected in a 1.5% agarose run with a 100 bp DNA ladder, a molecular weight standard. The number of tandem repeats was calculated based on the length of the repeat and flank sequences for each locus [13]. 

### 2.3. Data Analysis

The aggregated TB cases data at each province were used to calculate the TB incidence by dividing the total notification for each province for a particular year by the population of that given year, then multiplying by 100,000 [14]. 

Data management and analysis were performed using SPSS version-23 (SPSS Statistics 23, IBM: Corporation 1 New Orchard Road, Armonk, NY, USA). Factors associated with clustering, e.g., nationality, gender, province, and DST profiles were assessed by chi-square and Fisher’s exact tests. A *p*-value < 0.05 was considered as evidence of a significant difference.

Genetic clustering was defined as two or more isolates with identical spoligotypes. Recent transmission was estimated by calculating the clustering rate (CR) as: CR = (nc − c)/n; where nc = number of clustered isolates, c = number of clusters, and n = total number of isolates [15]. 

Binary logistic regression analysis was performed using R-packages software commands (version 1.4.1717) to find the association between the province of isolation and MTB sub-lineages. A statistical level of α < 0.05 was considered as evidence of significant association between variables. The strength of the association between MTB spoligotype and province was determined using the odds ratio and 95% confidence interval of the odds ratio.

A minimum spanning tree based on categorical distances was built by combining the spoligotyping and MIRU-VNTR results using MIRU-VNTR plus (https://www.miru-vntrplus.org/MIRU/index.faces, accessed on 21 March 2022), a freely accessible web-based program [16].

## 3. Results

### 3.1. Study Subjects

The present study examined 2,539 TB cases and the corresponding *M. tuberculosis* isolates in 9 provinces in Oman (North Al Batinah and South Al Batinah were grouped and the same was applied in regard to the Al Sharqiyah region) from nationals (1341 (52.8%)) and foreigners (1198 (47.2%)) between 2009 and 2018 (Figure 1). Most patients (837 (33.0%)) were aged 25–34, with a mean of 44.1 ± 20.2 and 33.7 ± 10.2 years among Omanis and foreigners, respectively. Most of the patients were male (n = 1549, 61.0%), with a slightly higher percentage of males among foreigners (62.3%) compared to Omanis (59.9%). The majority of the patients had pulmonary TB (2102, (82.8%)), while the remainder (437 (17.2%)) had extra-pulmonary TB (Appendix A).

### 3.2. Spatiotemporal Distribution of TB, 2009–2018, in Oman

Figure 2 shows the spatial distribution of TB cases in nine provinces between 2009 and 2018. Dhofar and Muscat had the highest rates of reported TB cases—8.4 and 8.2 per 100,000 population, respectively (Figure 1)—while Ad Dakhiliyah, Ash Sharqiyah, and Musandam showed low prevalences of 3.7, 2.9, and 2.5, respectively. Al Buraimi, Al Wusta, Al Batinah, and Adh Dhahira had intermediate prevalences of 5.8, 5.0, 5.1, and 4.1, respectively. This heterogeneous pattern of TB incidence is probably driven by socioeconomic and demographic factors.

Overall, TB cases in 2009 (7.9 per 100,000 population) declined steadily, reaching 4 per 100,000 in 2018, with an annual reduction of 45% (*p* < 0.05) (Figure 3A). The reduction was significant in Muscat (13.0 to 5.5 per 100,000, *p* = 0.002), Dhofar (10.4 to 5.1 per 100,000, *p* = 0.026), and Musandam (4.9 to 0.00/100,000), whereas TB rates remained largely remained unchanged in other provinces (Appendix A). A pronounced reduction in TB incidence was noted among Omanis (8.5 to 2.7 per 100,000) (*p* < 0.05) (Figure 3B), while the TB rate fluctuated among foreigners (6.8 to 5.8 per 100,000), with no consistent reduction (*p* = 0.126) (Figure 3B). 

### 3.3. Distribution of M. tuberculosis Lineages 

Spoligotyping of 1295 *M. tuberculosis* isolates collected from Omanis (n = 629) and foreigners (n = 666) in 9 provinces in Oman revealed 484 profiles, resulting in an overall diversity of 0.37 (Figure 1). Of these, 1030/1295 isolates (79.5%) matched the international database, while 265 isolates (20.5%) were ‘Orphan’. The known spoligotype profile comprised 13 sub-lineages: 6 major, including EAI (318, 30.9%), CAS (310, 30.1%), T (154, 14.9%), Beijing (88, 8.5%), LAM (64, 6.2%), and H (42, 4.1%); and 7 minor, including Ural (13, 1.3%), Manu (10, 1.0%), Cameroon (9, 0.9%), S (8, 0.8%), X (7, 0.7%), Turkey (5, 0.5%), and Zero (2, 0.2%).

No spatial variation was seen in the distribution of these sub-lineages in different provinces, with the exception of Beijing (*p* = 0.015, log likelihood = −274.63, χ2, 14.07) and CAS (*p* = 0.017, log likelihood = −632.73, χ2, 13.73) (Appendix A). Beijing was overrepresented in Dhofar and Ash Sharqiyah compared to Al Batinah, respectively (Dhofar: OR = 2.31, 95% CI, 1.13–4.79; Ash Sharqiyah: OR = 2.47, 95% CI, 1.07–5.59) (Appendix A), whereas CAS was 1.8 times and 2.0 times higher in Dhofar (OR = 1.88, 95% CI, 1.20–2.96) and Ad Dakhliyah (OR = 2.03, 95% CI, 1.17–3.49) compared to Al Batinah, respectively (Appendix A).

The major sub-lineages remained stable over the study period (Table 1), with the exception of EAI (*p* = 0.019, log likelihood = −706.25, χ2, 19.84) and Beijing (*p* = 0.022, log likelihood = −311.90, χ2, 19.33), which fluctuated significantly, with a lower probability of detecting EAI in 2017 compared to 2009 (OR 0.45, 95% CI, 0.26–0.77), while Beijing was 8.4 times, 5.0 times and 4.8 times higher in 2011, 2016, and 2017 compared to 2009, respectively (Table 1). A similar pattern was seen for the Orphan group (*p* = 0.002, log likelihood = −673.27, χ2, 25.95), with lower probability of detection in 2012 compared to 2009 (OR 0.22, 95% CI, 0.05–0.68) (Table 1).

### 3.4. Distribution of M. tuberculosis Lineages among Omanis and Foreigners

The above 13 sub-lineages were further divided into 45 different spoligo-clades of which 37 (82.2%) were shared between Omanis and foreigners (Appendix A). Of eight spoligo-clades, three (0.5%; EAI2, LAM5, and Manu3) were exclusive to Omanis and five (1.6%; EAI2-nonthaburi, LAM8, Manu1, Ural-1, and X1) were exclusive to foreigners (Appendix A). 

An association was seen between some spoligo-sub-lineages and nationality (Omani vs. foreigner), except for the T and LAM clades (*p* > 0.5). Beijing, H, and Orphan were significantly higher among foreigners (*p* < 0.05), while CAS and EAI were significantly higher among Omanis (*p* < 0.05) (Table 2).

### 3.5. Clustering of M. tuberculosis Lineages

The vast majority of the examined MTB isolates had shared spoligo-patterns (943/1296, 72.8%) and were grouped into 132 clusters of variable size (2 to 128 isolates per cluster), whereas 352 (27.2%) were distinctive. Of the 132 clusters, 85 were minor (2 to 4 isolates) and 47 were major spoligo-patterns (>5 isolates). The biggest clusters included CAS1_Delhi, SIT 26 (n = 128), Beijing SIT = 1 (n = 82), EAI5 SIT = 8 (n = 40), and CAS1_Delhi SIT = 25 (n = 40).

Clustering was not influenced by demographic variables, including patient gender, age, province, and DST profile (*p* > 0.05). A similar distribution of clustering was found among the mono (resistance to only one drug) (n = 125 (69.1%)), poly-resistant (resistance to more than one drug but not to both INH and RIF) (n = 20 (67%)), and MDR-TB isolates (n = 31 (88.6%)) (Table 3). However, clustering was affected by nationality, being higher among Omanis (74.4%) compared to foreigners (68.8%) (log likelihood= −770.24, χ2 = 5.35, *p =* 0.021) (Table 3), and spoligo-lineage, with the highest proportions of clustering in the Beijing (93.2%), CAS (89.7%), and EAI (83.8%) sub-linages. 

Interestingly, out of the 132 clusters, 77 (58.3%) were composed of *M. tuberculosis* lineages of both Omanis and foreigners (n = 770), whereas 23 (17.4%) clusters comprised only *M. tuberculosis* isolates from Omanis (n = 91) and 32 (24.2%) were composed exclusively of foreigners’ isolates (n = 82). 

### 3.6. MIRU-VNTR Analysis

A total of 192 *M. tuberculosis* isolates of 41 clusters were subjected to MIRU-VNTR analysis and 24-digit MIRU profiling to test whether these clusters consisted of identical genotypes, indicative of ongoing transmission, or if they were related but genetically distinct strains (Figure 1). 

Multi-locus genotypes (MLGs) of all isolates in the above clusters were highly diverse; each isolate carried a novel genotype. All of the above 41 spoligotype clusters were broken down into unique MIRU-VNTR genotypes (Appendix A), implying absence of recent transmission among the examined TB cases. 

## 4. Discussion

This study has revealed heterogeneous TB incidence in different provinces in Oman between 2009 and 2018, with a highly diverse reservoir of *M. tuberculosis* lineages comprising 45 clades in 1295 isolates. Most of the isolates (72.8%) were clustered in shared spoligotypes. The vast majority of the clusters (58.3%) comprised isolates of both Omanis and foreigners, indicative of close genetic relatedness between the *M. tuberculosis* lineages infecting the two groups. However, analysis of the major clusters did not detect any evidence of recent transmission. 

TB burden in Oman exhibits special heterogeneity, being disproportionately higher in Muscat (8.2 per 100,000 population) and Dhofar (8.4 per 100,000 population), respectively (Figure 2), and has been consistent across the study years (Appendix A). These findings are in line with a pervious analysis that revealed heterogeneity in TB incidence between 1981 and 2016, with the highest incidences in Muscat (105 per 100,000), North Batinah (an average of 63 per 100,000), and Dhofar (an average of 54 per 100,000) [5]. Muscat and Dhofar provinces have large foreign communities, which are associated with overcrowding of habitats and possibly heightened disease incidence. The large foreign communities, the majority of whom are from high-TB burden countries, can sustain latent TB reservoirs [17]. Dhofar and Muscat regions are also considered to have a high burden of multidrug-resistant TB (MDR-TB), defined as resistance to at least rifampicin (RIF) and isoniazid (INH) [18]. MDR strains have been linked to direct household transmission and clusters of recent transmission [19]. Additional characteristics associated with dissemination of the disease and high TB risk include demographic factors (e.g., living habitat), susceptibility to infection (immune status), and environmental factors—all major drivers of the heterogeneity of TB incidence [20]. A better insight into characteristics that drive high TB incidence in Oman would allow interventions targeting areas of high-risk groups. 

The present study reaffirms the decreased TB incidence over the study period (2009 to 2018) that has been reported before demonstrating the effectiveness of interventions to reduce TB transmission [5,6]. Nonetheless, a pronounced reduction in TB incidence was noted among Omanis (*p* < 0.05), while TB rates remained consistent among foreigners (*p* = 0.126) (Figure 2B). 

The major spoligotype sub-lineages, EAI (318, 30.9%), CAS (310, 30.1%), T (154, 14.9%), and Beijing (88, 8.5%), detected in the present study are common in the Indian subcontinent and East Africa. The above lineages are common in the Indian subcontinent (India, Pakistan, Bangladesh) and South East Asia (Indonesia, Philippines) [21,22,23,24]. Similarly, the T, LAM, and H lineages, which together account for 26.8% of the examined isolates in Oman, are common in Africa [25]. Again, this is in accordance with the fact that a large proportion of foreigners in Oman are from Africa. These lineages are common in East African countries, such as Tanzania, which has a historical link with Oman. The distribution of *M. tuberculosis* lineages in Oman mirrors that in Saudi Arabia, which has a large foreign community with similar demographic characteristics [26]. Similar to Oman, predominant *M. tuberculosis* sub-lineages in Saudi Arabia, Delhi/CAS, and Beijing, along with EAI sub-lineages, prevail in the Indian subcontinent and Southeast Asia [26]. The structure of *M. tuberculosis* sub-lineages in Oman and Saudi Arabia can also be attributed to the central geographical position of the Arabian Peninsula at the cross-roads between Asia and Africa and its ancient trade links with Southeast Asia and Africa [27,28,29]. 

The major spoligotype clades remained stable in space and time, with the exception of EAI (*p* = 0.019) and Beijing (*p* = 0.022), which fluctuated significantly over the study period (Table 1). The high diversity of *M. tuberculosis* in Oman and the stability of sub-lineages parallel those seen in areas of high TB burden [25].This suggests the presence of a large pathogen reservoir. The introduction of novel *M. tuberculosis* sub-lineages into the region via foreigners from high TB burden areas can enhance the diversity and effective population size (*Ne*), as there is a direct relationship between the expected level of diversity and *Ne* [30]. A large proportion of foreigners from high-TB burden countries may harbor the infection as latent TB. During the study period, 2009–2018, the population of foreigners has increased from 1156.358 to 2030.194 (NCSI. 2022) Thus, TB imported via foreigners represents not only a risk for augmenting local transmission but also the ability to disseminate novel strains that can escape the effects of current drug regimens.

An epidemiologically relevant finding was the high similarity of *M. tuberculosis* linages distributed among Omanis and foreigners (Appendix A). Out of 45 spoligotyping patterns (clades) detected, 37 (82.2%) were shared between the two groups, while only 3 (0.5%) clades (EAI2, LAM5, and Manu3) and 5 (1.1%) (EAI2-nonthaburi, LAM8, Manu1, Ural-1, and X1) were unique to Omanis and foreigners, respectively (Appendix A). These findings are in line with the findings from Saudi Arabia, where no differences were seen among phylogenetic lineages infecting nationals and foreigners [26]. However, the pattern seen in Oman and Saudi Arabia contrasts with that reported in areas with large immigrant communities in Europe and the US, where the Euro-American lineage strongly predominates among the isolates from native patients, while lineages associated with immigrants are rarely found among these patients [31,32]. Furthermore, mixed-cluster lineages (i.e., including isolates from both native and foreigners) constituted a large proportion of the total number of patient strain clusters in Oman (58.3%) and Saudi Arabia (59.5%) [26], while the above pattern of clustering has rarely been seen in Western countries [33,34,35]. The high rate of mixed-cluster phylogenetic lineages among nationals and foreigners in Oman and Saudi Arabia suggests a prevailing culture of close interaction that augments the probability of transmission between the two groups [36]. A large proportion of foreigners in Oman and Saudi Arabia occupy household jobs and are in daily contact with local families [26], whereas socio-economic and cultural barriers reduce contact between foreigners and native populations in Europe and US settings [37].

In the present study, the vast majority of examined *M. tuberculosis* isolates 943/1296 (72.8%) were grouped into 132 clusters of variable sizes, whereas 352 (27.2%) were distinctive. Of the 132 clusters, 85 were minor spoligo-patterns (2 to 4 isolates), while 47 were regarded as major spoligo-patterns (5 or more isolates). The biggest clusters included CAS1_Delhi, SIT 26 (n = 128), Beijing SIT = 1 (n = 82), EAI5 SIT = 8 (n = 40), and CAS1_Delhi SIT = 25 (n = 40). Analysis of correlations between clustering and demographic variables showed that clustering was not influenced by demographic variables, including patient gender, age, province, and DST profile (*p* > 0.05). The lack of association between clustering and *M. tuberculosis* drug response profile has previously been reported in other countries [36]; however, a comparatively higher clustering was seen among Omanis (74.4%) compared to foreigners (68.8%) (*p =* 0.021) (Table 3). These findings are consistent with those among TB patients in Saudi Arabia indicating that nationals had a higher rate of clustering (50.3%) compared to non-Saudi patients (45.8%) (Varghese et al., 2013). This contrasts with the pattern of clustering in Western countries, where limited shared clustering is generally found between native and immigrants TB populations [37].

We further analyzed *M. tuberculosis* in large clusters in Oman, using standard 24-locus MIRU-VNTR typing, known to provide superior discriminatory power, to test the hypothesis that the persistence of TB in Oman is driven by transmission between foreigners and Omanis [6,38]. All of the 192 isolates, in 41 clusters, were found to have a unique combination of the examined MIRU-VNTR loci, hence there was no evidence of transmission between Omanis and foreigners or among any of the two groups. Thus, in the case of Oman, MIRU_VNTR offers an advantage over spoligotyping for genotyping *M. tuberculosis* to assess areas of clustering and active transmission for targeted control. Nonetheless, whole genome sequencing, which has a much higher discriminatory power than MIRU-VNTR, can be considered for clinical strains exhibiting identical MIRU-VNTR genotyping profiles for better interpretation of transmission patterns. 

In summary, the present study has revealed heterogeneous spatial TB risk in Oman and a high extent of genetic diversity in *M. tuberculosis*, consistent across the ten years of the study period. Nonetheless, there is high similarity between *M. tuberculosis* lineages infecting Omanis and foreigners and a high proportion of shared clustered lineages containing isolates of the two groups, indicative of close social interaction favorable for TB transmission. Nonetheless, no evidence of ongoing transmission was evident from MIRU-VNTR alleles. Further analysis of recent incidences in high-TB risk areas, in Muscat and Dhofar, using highly discriminatory markers, MIRU-VNTR or whole genome, is critical for targeted intervention.

## Figures and Tables

**Figure 1 pathogens-11-00541-f001:**
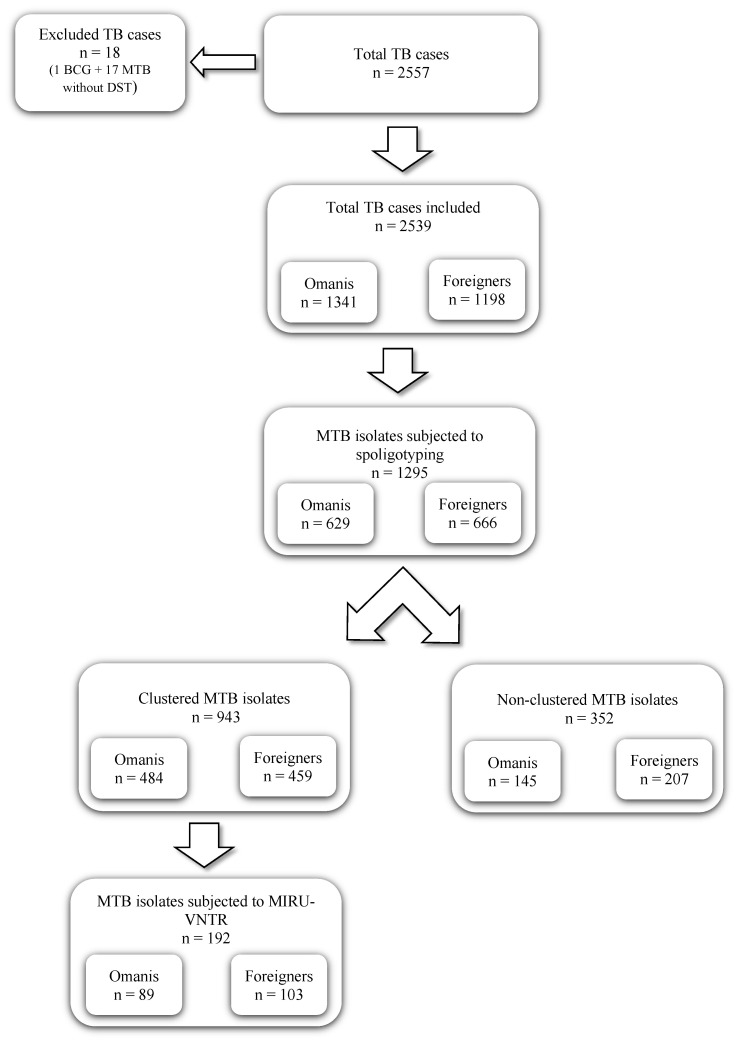
Flowchart of the workflow analysis applied for MTB isolates.

**Figure 2 pathogens-11-00541-f002:**
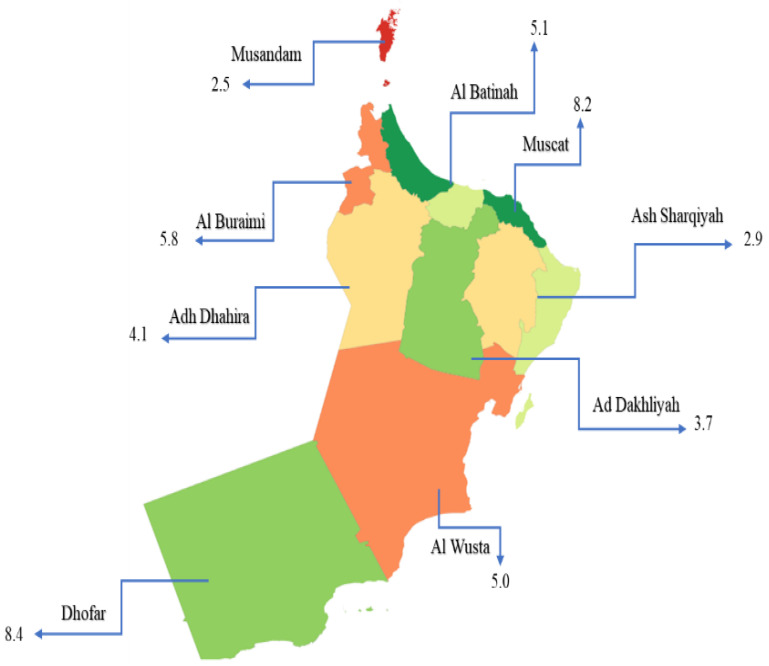
Spatial distribution of TB cases per 100,000 of the population in different provinces in Oman between 2009 and 2018. North and South Al Batinah were grouped as one province (Al Batinah); similarly, South and North Ash Sharqiyah were grouped as one province (Ash Sharqiyah).

**Figure 3 pathogens-11-00541-f003:**
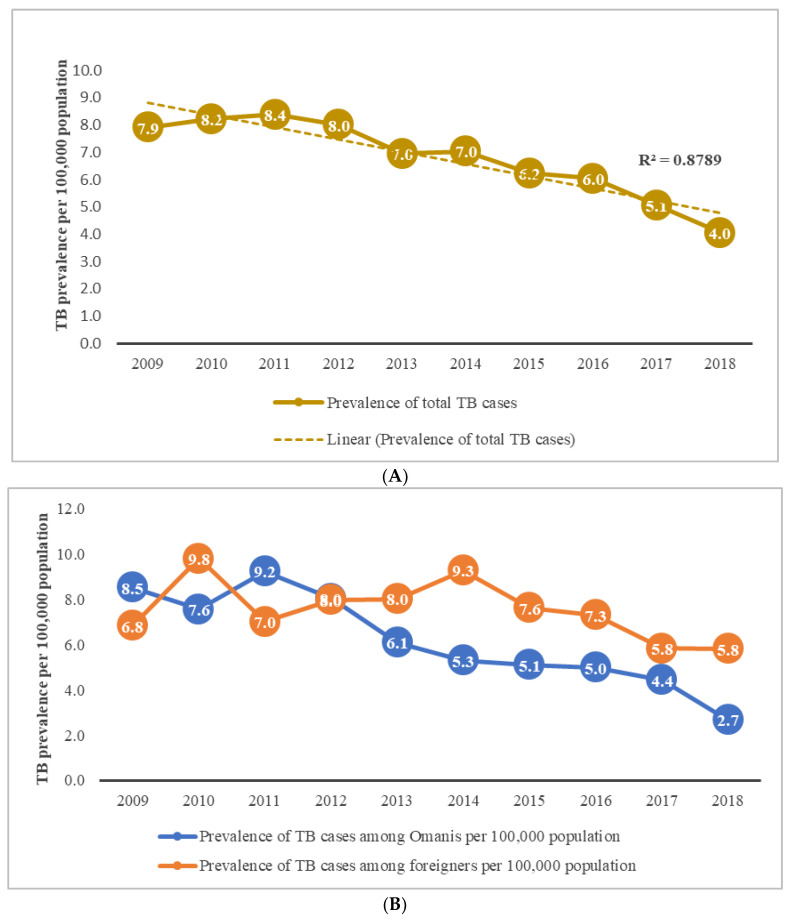
Prevalence of total TB cases per 10,000 population in Oman between 2009 and 2018: (**A**) overall pattern; (**B**) prevalence among Omanis and foreigners.

**Table 1 pathogens-11-00541-t001:** Distribution of *M. tuberculosis* lineages in Oman between 2009 and 2018.

	Total	2009 n = 136	2010 n = 35	2011 n = 50	2012 n = 56	2013 n = 255	2014 n = 156	2015 n = 150	2016 n = 148	2017 n = 195	2018 n = 114	*p*-Value
EAI	318 (30.9%)	44 (32.4%)	7 (20.0%)	9 (18.0%)	22 (39.3%)	61 (23.9%)	39 (25.0%)	41 (27.3%)	41 (27.7%)	31 (15.9%)	23 (20.2%)	0.019
Beijing	88 (8.5%)	3 (2.2%)	3 (8.6%)	8 (16.0%)	4 (7.1%)	14 (5.5%)	9 (5.8%)	9 (6.0%)	15 (10.1%)	19 (9.7%)	4 (3.5%)	0.022
CAS	310 (30.1%)	32 (23.5%)	7 (20.0%)	11 (22.0%)	18 (32.1%)	67 (26.3%)	39 (25.0%)	37 (24.7%)	28 (18.9%)	50 (25.6%)	21 (18.4%)	0.629
T	154 (14.9%)	12 (8.8%)	6 (17.1%)	6 (12.0%)	4 (7.1%)	28 (11.0%)	22 (14.1%)	21 (14.0%)	18 (12.2%)	20 (10.3%)	17 (14.9%)	0.613
LAM	64 (6.2%)	15 (11.0%)	1 (2.9%)	3 (6.0%)	3 (5.4%)	13 (5.1%)	7 (4.5%)	6 (4.0%)	5 (3.4%)	6 (3.1%)	5 (4.4%)	0.585
H	42 (4.1%)	4 (2.9%)	2 (5.7%)	0	2 (3.6%)	13 (5.1%)	4 (2.6%)	4 (2.7%)	6 (4.1%)	5 (2.6%)	2 (1.8%)	0.320
Others ^1^	54 (4.2%)	6 (4.4%)	1 (2.9%)	2 (4.0%)	1 (1.8%)	9 (3.5%)	5 (3.2%)	8 (5.3%)	3 (2.0%)	10 (5.1%)	9 (7.9%)	0.011
Orphan	265 (20.5%)	20 (14.7%)	8 (22.9%)	11 (22.0%)	2 (3.6%)	50 (19.6%)	31 (19.9%)	24 (16.0%)	32 (21.6%)	54 (27.7%)	33 (28.9%)	0.002

^1^ Minor lineages (Ural (13), Manu (n = 10), X (n = 7), S (n = 8), Cameroon (n = 9), Turkey (n = 5), Zero (n = 2)).

**Table 2 pathogens-11-00541-t002:** *M. tuberculosis* clades among Omanis and foreigners.

	Foreigners n (%)	Omani n (%)	Total	*p*-Value *
EAI	145 (21.8%)	173 (27.5%)	318 (24.5%)	0.017
Beijing	56 (8.4%)	32 (5.1%)	88 (6.8%)	0.018
CAS	133 (20.0%)	177 (28.1%)	310 (23.9%)	0.001
T	78 (11.7%)	76 (12.1%)	154 (11.9%)	0.837
H	31 (4.7%)	11 (1.7%)	42 (3.2%)	0.003
LAM	28 (4.2%)	36 (5.7%)	64 (4.9%)	0.207
Others	40 (6.0%)	14 (2.2%)	54 (4.2%)	0.001
Orphan	155 (23.3%)	110 (17.5%)	265 (20.5%)	0.010
Total	666	629	1295	

* Chi-square.

**Table 3 pathogens-11-00541-t003:** Effect of demographic variable on the clustering of *M. tuberculosis* lineages.

Variable		Clustered	Non-Clustered	*p*-Value
		n (%)	n (%)	
Year	2009	73 (55.7)	58 (44.3)	0.385
	2010	6 (17.1)	29 (82.9)	
	2011	20 (40.0)	30 (60.0)	
	2012	25 (44.6)	31 (55.4)	
	2013	151 (59.2)	104 (40.8)	
	2014	80 (53.7)	69 (46.3)	
	2015	95 (65.5)	50 (34.5)	
	2016	73 (56.1)	57 (43.9)	
	2017	87 (50.6)	85 (49.4)	
	2018	42 (40.0%)	63 (60.0%)	
Sex	Male	556 (70.4)	234 (29.6)	0.279
	Female	367 (73.3)	134 (26.7)	
Nationality	Omani	469 (74.4)	160 (31.2)	0.021
	Foreigners	458 (68.8)	208 (25.6)	
Age group	Child	9 (52.9)	8 (47.1)	0.174
	Adult	919 (71.8)	366 (28.5)	
Area of residence	Muscat	375 (68.6)	172 (31.4)	0.300
	Al Batinah	206 (74.9)	69 (25.1)	
	Dhofar	118 (77.1)	35 (22.9)	
	Al Dakhilia	59 (71.1)	24 (28.9)	
	Al Sharqiya	62 (70.5)	26 (29.5)	
	Al Buraimi	19 (61.3)	12 (38.7)	
	Al Wusta	7 (87.5)	1 (12.5)	
	Al Dhahira	35 (76.1)	11 (23.9)	
	Musandam	4 (80.0)	1 (20.0)	
DST profiles	Sensitive	747 (71.6)	297 (28.4)	0.078
	MonoR	125 (69.1)	56 (30.9)	
	PolyR	20 (76.9)	6 (23.1)	
	MDR	31 (88.6)	4 (11.4)	
Sub-lineages	CAS	288 (89.7)	33 (10.3)	<0.05
	EAI	274 (83.8)	53 (16.2)	
	Beijing	82 (93.2)	6 (6.8)	
	H	26 (61.9)	16 (38.1)	
	LAM	54 (77.1)	16 (22.9)	

## Data Availability

The authors confirm that the data supporting the findings of this study are available within the article if required.

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
