# Peer review of "Dynamics of Mycobacterium tuberculosis Lineages in Oman, 2009 to 2018"

_pathogens, 2022, doi:10.3390/pathogens11050541_

Round 1

Reviewer 1 Report

Overall Comments to Author(s)
==============================
The manuscript by Sara Al-Mahrouqi et al titled "Dynamics of Mycobacterium tuberculosis lineages in Oman, 2008 to 2018" 
describes the analysis of TB patient demographics, comparing between locals (Omanis) and non-locals, examining the genetic relatedness 
of the TB bacteria as assessed through spoligotyping and MIRU-VNTR. 
Some clustering were identified through spoligtypes but further analysis through MIRU-VNTR did not detect recent on-going transmission.

The results should be published, and made available such as through submission to SITVIT2, to provide a deeper understanding and monitoring 
of the TB molecular epidemiology globally and regionally. 

There are however several suggestions that the authors should consider in improving the current version of the manuscript: 
1. The title or period of study: Should it be 2009 to 2018?
2. Further elaboration on the study design and methods would be helpful to aid readers in understanding how the analysis were performed. 
   Perhaps a flowchart depicting the analysis workflow should be considered for inclusion. 
3. Breakdown of non-locals into countries of origin and inclusion for analysis would greatly improve resolution of results.
4. Please check the numbers/values as indicated in the manuscript for their accuracies. 

Major Comments
---------------
Line 18: .... migration from high TB burden "countries". The word "countries" seem to be missing

Line 20: "TB cases reported between 2009 to 2018.. ". 
The title of the manuscript however states 2008 to 2018.
There are multiple instances where for example Table 1, the title states between 2008 and 2018, but the values in the table displays 2009 onwards.
From the occurrences, it might seem that the title should be updated to better reflect the results collected, i.e. 2009 to 2018.

Line 22: "A total of 482 spoligotypes were detected among the 22 examined isolates, 943 (72.8%) shared 132 clusters while the remaining 352 (27.2%) each had a 23 unique pattern."
Do the numbers 943 and 352 refer to distinct isolates/samples or spoligotypes? If its the latter, it does not add to 482 however.
Similarly, the addition of 132 clusters and 352 unique patterns also does not add up to 482 spoligotypes.

Line 24: "The most frequent spoligotypes belong to EAI (318, 30.9%), CAS (310, 30.1%), T 24 (154, 14.9%), and Beijing (88, 8.5%) lineages."
There needs to be clear distinction about whether you are describing isolates/TB cases versus spoligotypes as the numbers are very confusing. 
Multiple TB cases/isolates can share one spoligtype pattern.

Lines 22-26: The results breakdown should also be by nationality, i.e. Omanis and expatriates. 
For example 318 TB cases (if it's interpreted correctly) with spoligotypes that belong to EAI, how many are Omanis and expatriates respectively?
Also, it might be better to use foreigners or non-locals or some other terms instead of expatriates. 

Lines 27-30: The statements made in the results and conclusion sections are conflicting. 
In results section, it states no evidence of ongoing transmission was found and 
yet in the conclusion, it was mentioned that the pattern of sptial heterogenetity across Oman reflects social links that favor transmission between the two groups.
The authors should be consistent and arrive at one conclusion supported by results and evidence.

Lines 55-56: "... 1981-1992, .. decreased by 11% .. from 91 to 37 .. "
             " between 1992 and 2016 ... decrease from 21 to 14 cases ...  "
How is it that the  incidence rate for 1992 is both 37 and 21?

Line 105: Correct url for SITVIT2 database should be http://www.pasteur-guadeloupe.fr:8081/SITVIT2/
The url stated in the current manuscript version is invalid.

Lines 109-110: How are the 192 Mtb samples selected for MIRU-VNTR? 
How do you select these 41 spoligo lineages amongst the 132 clusters as stated in the Abstract line 23?  

Lines 135-136: Correct url for MIRU-VNTR plus website should be https://www.miru-vntrplus.org/MIRU/index.faces

Figure 1: Please state the legend explicitly
Does the color encoding represent anything? The rates indicated are mean rates throughout the years assessed?

Lines 157-158: Please provide supporting evidence for this statement.

Lines 162-170: Please check your numbers. P=0.000? While the P-values can be signficant, it cannot be zero. 
               ... Musandam (4.9 to 0.00/105)? Musandam is 2.5 per 100,000 as reflected in Figure 1.

Lines 176: "Spoligotyping of 1,295 ... " 
It is not mentioned in the Methods section that only a subset of isolates were spoligotyped instead of the full dataset of 2,539. 
It should be stated why and how this subset of samples were subjected to spoligotyping. 

Line 187: "95%CI, 01.13-4.79 ... ". it should be stated as 1.13-4.79.  

Table 2: Percentages for Muscat should be enclosed in brackets, similar to other provinces. 

Lines 195-199: " ... with a lower risk of detecting EAI ... ". 
Can you please elaborate what's the risk of detecting EAI and Beijing?

Lines 226: Please define what is meant by mono and poly-resistant.

Lines 258-260: MIRU-VNTR analysis did not detect any evidence of recent transmission though  (lines 251-252), so how is it that there can possibly be heightened disease transmission?

Lines 276-292: Can the results displayed be stratified into the expatriates' country of origin? 
Or at least having certain distribution as currently there are no distribution of such demographics information in the manuscript.

Lines 297-303: Has there been sustained or increased migration of expatriates into the country?

Minor Comments
--------------
The font formatting of the words in the Abstract seem different for the different headings.

Line 65: " ... of expatriates’ form countries with ... "
form -> from?

Line 244: transition -> transmission?

Supplementary Figure 1: Please ensure consistent font sizes. What do the xxx mean?

Supplementary Figure 2: Please include legends. 
The first block at the top should represent spoligotypes, while the next block of numbers MIRU-VNTR genotypes (which should clearly indicate the order of the loci).
Please clearly indicate the method of tree building.

Supplementary Table 1: Last column header should be included, i.e. "Total"

Author Response

We thank the reviewers for their constructive and helpful comments. We addressed all of the comments and detailed our response below. In addition, the manuscript has been edited and check throughout, for grammar and typos.

Reviewer 1.

The manuscript by Sara Al-Mahrouqi et al titled "Dynamics of Mycobacterium tuberculosis lineages in Oman, 2008 to 2018"  describes the analysis of TB patient demographics, comparing between locals (Omanis) and non-locals, examining the genetic relatedness of the TB bacteria as assessed through spoligotyping and MIRU-VNTR.  Some clustering were identified through spoligtypes but further analysis through MIRU-VNTR did not detect recent on-going transmission.

The results should be published, and made available such as through submission to SITVIT2, to provide a deeper understanding and monitoring of the TB molecular epidemiology globally and regionally. 

Response: We are pleased that the reviewer finds the manuscript of interest and relevant to understanding of regional and global TB epidemiology

There are however several suggestions that the authors should consider in improving the current version of the manuscript: 
1. The title or period of study: Should it be 2009 to 2018?

Response: We thank the reviewer; the tittle has been edited

  1. Further elaboration on the study design and methods would be helpful to aid readers in understanding how the analysis were performed. 
     Perhaps a flowchart depicting the analysis workflow should be considered for inclusion. 

Response: A flowchart showing workflow of the analysis has been introduced (Figure 1). 

  1. Breakdown of non-locals into countries of origin and inclusion for analysis would greatly improve resolution of results.

Response: We agree with reviewer that this is an important information to add, however, unfortunately this data is not accessible. 

  1. Please check the numbers/values as indicated in the manuscript for their accuracies. 

Response: We thank the reviewer, the numbers and values have been checked and corrected throughout the text.

Major Comments
---------------
Line 18: .... migration from high TB burden "countries". The word "countries" seem to be missing

Response: We thanks the reviewer for this, the sentence has been corrected

Line 20: "TB cases reported between 2009 to 2018.. ". 

Response: These are the correct dates of the studyThe title of the manuscript however states 2008 to 2018.
There are multiple instances where for example Table 1, the title states between 2008 and 2018, but the values in the table displays 2009 onwards.
From the occurrences, it might seem that the title should be updated to better reflect the results collected, i.e. 2009 to 2018.

Response: We thank the reviewer, the title has been edited to reflect the correct dates of the study, 2009-2018

Line 22: "A total of 482 spoligotypes were detected among the examined isolates, 943 (72.8%) shared 132 clusters while the remaining 352 (27.2%) each had a unique pattern."
Do the numbers 943 and 352 refer to distinct isolates/samples or spoligotypes? If its the latter, it does not add to 482 however.
Similarly, the addition of 132 clusters and 352 unique patterns also does not add up to 482 spoligotypes.

Response: The number throughout the study indicates, unique M. tuberculosis isolates, collected from individual patients. Thus, the 943 and 352 add up to 1295 (total number examined). We have added a flowchart to clarify the number of isolates at different stage of the study. With regard to total number of spoligotypes, the correct number was 484, where 943 MTB isolates were found in 132 clusters and the remaining are found as singletons (unclusteres, n=352 MTB isolates). Adding up 132plus 352 gives 484 spoligotypes.

Line 24: "The most frequent spoligotypes belong to EAI (318, 30.9%), CAS (310, 30.1%), T 24 (154, 14.9%), and Beijing (88, 8.5%) lineages."    There needs to be clear distinction about whether you are describing isolates/TB cases versus spoligotypes as the numbers are very confusing. Multiple TB cases/isolates can share one spoligtype pattern.

Response: The frequency of a spoligotype was calculated as the proportion of all isolates carrying the spoligotype out of all examined successfully tested. While, isolates refer to an individual samples of MTB taken from one patient.

Lines 22-26: The results breakdown should also be by nationality, i.e. Omanis and expatriates. 
For example 318 TB cases (if it's interpreted correctly) with spoligotypes that belong to EAI, how many are Omanis and expatriates respectively?
Also, it might be better to use foreigners or non-locals or some other terms instead of expatriates. 

Response: We added a sentence describing that “there were differences in the frequencies of the major spoloigotype of MTB isolates infecting Omanis and expatriates (foreigners)”.  The details of the frequencies of each lineage among the two groups are given in the results section (Table 4). 

We replaced the word “expatriates” with “foreigners” throughout the manuscript

Lines 27-30: The statements made in the results and conclusion sections are conflicting. 
In results section, it states no evidence of ongoing transmission was found and 
yet in the conclusion, it was mentioned that the pattern of sptial heterogenetity across Oman reflects social links that favor transmission between the two groups.
The authors should be consistent and arrive at one conclusion supported by results and evidence.

Response: We thank the reviewer for this comment, we edited the last sentence in the abstract to reflect the fact that we did not detect evidence for recent transmission between the two groups.   

Lines 55-56: "... 1981-1992, .. decreased by 11% .. from 91 to 37 .. "
             " between 1992 and 2016 ... decrease from 21 to 14 cases ...  "
How is it that the  incidence rate for 1992 is both 37 and 21?

Response: We have edited this section, to clearly explain the pattern of TB incidence between 1981 to 2016,  and corrected the information on TB rates between 1992-2016, Lines 58-60

Line 105: Correct url for SITVIT2 database should be http://www.pasteur-guadeloupe.fr:8081/SITVIT2/
The url stated in the current manuscript version is invalid.

Response: We thank the reviewer for these suggestions, we have updated the url

Lines 109-110: How are the 192 Mtb samples selected for MIRU-VNTR? 
How do you select these 41 spoligo lineages amongst the 132 clusters as stated in the Abstract line 23?

Response: We have randomly selected samples in the major of spoligotypes clusters, from different provinces.  As mentioned above we have added a flowchart to clarify the number of samples examined at different stages of the study.  

Lines 135-136: Correct url for MIRU-VNTR plus website should be https://www.miru-vntrplus.org/MIRU/index.faces

Response:  We thank the reviewer for these suggestions, we have updated the url

Figure 1: Please state the legend explicitly.

Response: Extra information was added to explain that four adjacent provinces were grouped into two provinces. North & South Al Batinah (Al Batinah), and North & South Ash Sharqiyah (Ash Sharqiyah)

Does the color encoding represent anything? The rates indicated are mean rates throughout the years assessed?

Response: The colors, designate different provinces in Oman, and do not reflect disease incidence.

Lines 157-158: Please provide supporting evidence for this statement.

Response: The sentence was deleted

Lines 162-170: Please check your numbers. P=0.000? While the P-values can be significant, it cannot be zero. 
               ... Musandam (4.9 to 0.00/105)? Musandam is 2.5 per 100,000 as reflected in Figure 1.

Response: We thank the reviewer , the P = 0.000 has been changed to  P < 0.05.  

Regarding TB rate in Musandam, Figure 1 indicates the overall prevalence of TB during the study period, while Table 1, shows yearly Tb rate per 105 in  Musandam (4.9 to 0.00/105) over the study period, with a mean of 2.5 per 105.

Lines 176: "Spoligotyping of 1,295 ... " 
It is not mentioned in the Methods section that only a subset of isolates were spoligotyped instead of the full dataset of 2,539. 
It should be stated why and how this subset of samples were subjected to spoligotyping. 

Response: Spoligotyping was carried out on successfully cultured MTB isolates.  A flowchart was added to reflect the samples analysed at different stages of the study

Line 187: "95%CI, 01.13-4.79 ... ". it should be stated as 1.13-4.79.

Response: Corrected  

Table 2: Percentages for Muscat should be enclosed in brackets, similar to other provinces. 

Response: Corrected

Lines 195-199: " ... with a lower risk of detecting EAI ... ". 
Can you please elaborate what's the risk of detecting EAI and Beijing?

Response: We thank the reviewer for raising this, we have now edited this sentence to reflect the probability of detection of each lineage (e.g. lines 230-237).

Lines 226: Please define what is meant by mono and poly-resistant.

Response: The terms mon and poly-resistant have been explained, page 12, lines-274-75

Provide this in the

Lines 258-260: MIRU-VNTR analysis did not detect any evidence of recent transmission though  (lines 251-252), so how is it that there can possibly be heightened disease transmission?

Response: We replaced the word “transmission” with “incidence”.

Lines 276-292: Can the results displayed be stratified into the expatriates' country of origin? 
Or at least having certain distribution as currently there are no distribution of such demographics information in the manuscript.

Response: Unfortunately, this information is not accessible.

Lines 297-303: Has there been sustained or increased migration of expatriates into the country?

 Response: The expatriates population has increased during the study period, from 1.156.358 in 2009 to 2030.194 in Dec 2018. We added this information in the discussion section, page 14 para 4.

Minor Comments
--------------
The font formatting of the words in the Abstract seem different for the different headings.

 Response: Corrected

Line 65: " ... of expatriates’ form countries with ... "
form -> from?

Response: Corrected

Line 244: transition -> transmission?

Response: Corrected

Supplementary Figure 1: Please ensure consistent font sizes. What do the xxx mean?

Response: Corrected, xxx was replaced with the defined sub-lineages

Supplementary Figure 2: Please include legends. 
The first block at the top should represent spoligotypes, while the next block of numbers MIRU-VNTR genotypes (which should clearly indicate the order of the loci).
Please clearly indicate the method of tree building.

Response: Corrected

Supplementary Table 1: Last column header should be included, i.e. "Tota

Response: Corrected

Reviewer 2 Report

The study describes the TB incidence and the TB strain diversity between 2009 and 2018 in Oman.

Comments:

  1. The introduction (lines 53-62) describes the TB incidence in Oman for the last 25 years. For the time period of 1992 to 2016 a total number of 2352 TB cases is named. This number seems too low considering the incidence (21-14 for the years 1992-2016) and population of Oman. Please explain and specify the time period and numbers. Also, for the year 1992 different TB incidences are mentioned (37 vs. 21). In addition, there is a discrepancy between the TB incidence described in the introduction and shown in Figure 2 for the same period of time. Please explain.
  2. Line 92. The study has analysed 2539 TB cases of the years 2009-2018. Is this the total number of TB cases in Oman for this period of time? Please specify.
  3. Material and methods. Lines 99-114. Spolitotyping was done for 1295 of 2539. It should be outlined here which strains were selected for spoligotyping. Selection bias? Same for MIRU typing, only a subset of strains was analysed. Why? Explain the selection process.
  4. Line 146. Refers to suppl. table 1 for data on pulmonary vs. non-pulmonary TB. However, these data are not shown in suppl. table 1.
  5. Line 151. Corret time period 2009-2018
  6. Line 140 and Line 152. Data shown are from 9 provinces of Oman as described in the manuscript. Do these 9 provinces cover the whole country? Are there additional provinces not included in the study? Compare also figure 1 which shows 11 differently colored regions? Please explain.
  7. Lines 168-170. Delete this statement or move to the discussion.
  8. Spoligotyping (lines 175ff). Numbers in the text, table 2 and table 3 are not consistent. E.g. text EAI n=318, table 2 EAI n=288, and table 3 EAI n=313. Please check the numbers.
  9. Lines 206-207. Delete this statement or move to the discussion.
  10. Line 209. (P>0.05)
  11. Line 218. What was the definition of «similar spoligo-pattern»?

Tables and figures. There are too many figures and tables with repeating content in the main manuscript and supplementary data.

Table 1. Move to suppl. data

Table 2. Move to suppl. data

Figure 1. Indicate the meaning of the colors. Figure 1 shows 11 differently colored regions; but the text talkes about 9 provinces? Explain.

Figure 2. Fig 2A and Fig 2B show comparable data but the graphical representation is different. This is confusion. Please merge Fig 2A and 2B in one single graph or at least unify the graphs.

Figure 3. Delete or move to suppl. data.

Suppl. Figure 1. Delete. Repetition.

Author Response

We thank the reviewers for their constructive and helpful comments. We addressed all of the comments and detailed our response below. In addition, the manuscript has been edited and check throughout, for grammar and typos.

Reviewer 2.

The study describes the TB incidence and the TB strain diversity between 2009 and 2018 in Oman.

Comments:

  1. The introduction (lines 53-62) describes the TB incidence in Oman for the last 25 years. For the time period of 1992 to 2016 a total number of 2352 TB cases is named. This number seems too low considering the incidence (21-14 for the years 1992-2016) and population of Oman. Please explain and specify the time period and numbers. Also, for the year 1992 different TB incidences are mentioned (37 vs. 21). In addition, there is a discrepancy between the TB incidence described in the introduction and shown in Figure 2 for the same period of time. Please explain.

Response: We have edited this section and corrected the information on TB rates between 1992-2016, Lines 58-60

  1. Line 92. The study has analysed 2539 TB cases of the years 2009-2018. Is this the total number of TB cases in Oman for this period of time? Please specify.

Response: The study included all MTB samples with in vitro DST profiles, we excluded M. bovis and samples with no DST data.

  1. Material and methods. Lines 99-114. Spolitotyping was done for 1295 of 2539. It should be outlined here which strains were selected for spoligotyping. Selection bias? Same for MIRU typing, only a subset of strains was analysed. Why? Explain the selection process.

Response: A flowchart showing workflow of the analysis has been introduced.  The samples subjected to spoligotyping were randomly selected based availability of material for DNA extraction. With regards to MIRU typing, only isolates with shared spoligotype profiles (clustered), and available material, were examined.

  1. Line 146. Refers to suppl. table 1 for data on pulmonary vs. non-pulmonary TB. However, these data are not shown in suppl. table 1.

Response: Suppl. Table 1 was edited, and the information has been added.

  1. Line 151. Corret time period 2009-2018

Response: We thank the reviewer for noting the above errors. The study period dates have been corrected throughout the manuscript.

  1. Line 140 and Line 152. Data shown are from 9 provinces of Oman as described in the manuscript. Do these 9 provinces cover the whole country? Are there additional provinces not included in the study? Compare also figure 1 which shows 11 differently colored regions? Please explain.

Response: Extra information was added to explain that four adjacent provinces were grouped into two provinces. North & South Al Batinah (Al Batinah), and North & South Ash Sharqiyah (Ash Sharqiyah)

  1. Lines 168-170. Delete this statement or move to the discussion.

Response: The sentence was deleted

  1. Spoligotyping (lines 175ff). Numbers in the text, table 2 and table 3 are not consistent. E.g. text EAI n=318, table 2 EAI n=288, and table 3 EAI n=313. Please check the numbers.

Response: We thank the reviewer for noticing these inconsistences, these have been corrected.

  1. Lines 206-207. Delete this statement or move to the discussion.

Response: The sentence was deleted

  1. Line 209. (P>0.05)

Response: This P value indicates lack of association nationality and T and LAM clades (P>0.5). This value has been checked.

  1. Line 218. What was the definition of «similar spoligo-pattern»?

Response: The word “similar” was replaced with “shared”.

 Tables and figures. There are too many figures and tables with repeating content in the main manuscript and supplementary data.

Table 1. Move to suppl. data

Table 2. Move to suppl. Data

Response: We thank the reviewer for these suggestions, we moved Table 1 and Table 2 to supplementary material.  

Figure 1. Indicate the meaning of the colors. Figure 1 shows 11 differently colored regions; but the text talkes about 9 provinces? Explain.

Response: As mentioned above (response to point 6), adjacent provinces were grouped into two provinces. North & South Al Batinah (Al Batinah), and North & South Ash Sharqiyah (Ash Sharqiyah). Extra information was added to explain the above.

The colors, designate different provinces in Oman, and do not reflect disease incidence.

Figure 2. Fig 2A and Fig 2B show comparable data but the graphical representation is different. This is confusion. Please merge Fig 2A and 2B in one single graph or at least unify the graphs.

Response: The two figures have been combined into one.

Figure 3. Delete or move to suppl. data.

Response: Figure 3 has been changed to supplementary figure 1

Suppl. Figure 1. Delete. Repetition.

Response: Suppl. Figure 1 was deleted

Round 2

Reviewer 1 Report

Overall Comments to Author(s)
==============================
The current version of the manuscript has been signficantly improved, 
but there remain certains areas that could be further improved as highlighted in the following comments. 

There are multiple instances where the authors used 105 which is to mean 10 to the power of 5, where 5 is supposed to be superscript. 
It would have been better to simply use the number value 100,000 instead.
Here, the number 105 would have meant very differently (like in line 17 and supplementary table 2).
Some of the values appear correct in the manuscript such as in lines 45, 47..,
but it would probably have been much better to simply use 100,000 where a mistake in formatting would cause signficant 
Please correct these numbers in the manuscript, which are in 
Line 17: " ... <1 TB case per 105 population." 
Line 59, 60: " ... per 105"
Supplementary Table 2: 105 -> 100,000?

Line 23: "A total of 4824 spoligotypes ..."
Please correct number to 484.

Lines 29-33: 
I believe the authors wish to convey that spoligotype lineages are diverse and yet about half (77 out of 132 clusters) are shared between Omanis and foreigners, 
as illustrated in Lines 26-27: " the frequencies of these lineages differed between Omanis and foreigners." 
The use of MIRU-VNTR on some of the shared spoligotype clusters identify the clustered isolates as distinct and thus not having any evidence of recent transmission.
This is however not well-conveyed through the wordings currently used. 
In particular, line 29 states this may reflect transmission, while line 30-31 states no evidence of no-going transmission was found.
In addition,the amendments by the authors still states that there is spatial heterogeneity with high diversity and yet shared sub-lineages. 
This is confusing as if the lineages are diverse and heterogeneous, how is it shared?

The newly provided Figure 1 has signficantly improved the understanding of the samples collection, and procedures performed in this study.
From Figure 1, it is evident that about half (47% Omanis, and 56% foreigners) of all TB cases were successfully genotyped. 
Is there a particular reason why there is about 10% difference in genotyping success rate between Omanis and foreigners? 
Also, is there is a difference in demographic characteristics of those who were genoytped versus those not genotyped that could confound the clustering results? 

The information that spoligotyping was carried out on only successfully cultured MTB isolates, and MIRU-VNTR genotyping was performed on randomly selected samples in the major spoligotype clusters should be explictly stated in the Methods section.

Table 3: P-value for Sub-lineages is not possible to be 0.000.
Please correct.

Minor Comments
--------------
Line 61: "Between 2010 to 2016,, ..."
Remove extra comma

Line 197: "CI, 01.20-2.96: should be "1.20-2.96". 

Line 243: vatiable -> variable

Author Response

The current version of the manuscript has been signficantly improved, 
but there remain certains areas that could be further improved as highlighted in the following comments. 

Response: We are pleased that the reviewer finds the revised version significantly improved. We have revised the current version of the manuscript (in light of comments made by both reviewers), which has enabled us to put the interpretation of our results into a clearer, more accurate context.

There are multiple instances where the authors used 105 which is to mean 10 to the power of 5, where 5 is supposed to be superscript. 
It would have been better to simply use the number value 100,000 instead.
Here, the number 105 would have meant very differently (like in line 17 and supplementary table 2). Some of the values appear correct in the manuscript such as in lines 45, 47..,
but it would probably have been much better to simply use 100,000 where a mistake in formatting would cause signficant 
Please correct these numbers in the manuscript, which are in 
Line 17: " ... <1 TB case per 105 population." 
Line 59, 60: " ... per 105"
Supplementary Table 2: 105 -> 100,000?

Response: We thank the reviewer for this comment, the value “105” has been changed to “100,000” throughout the text as well as supplementary Table 2.

Line 23: "A total of 4824 spoligotypes ..."
Please correct number to 484.

Response: We thank the reviewer; this has been corrected

Lines 29-33: 
I believe the authors wish to convey that spoligotype lineages are diverse and yet about half (77 out of 132 clusters) are shared between Omanis and foreigners, as illustrated in Lines 26-27: " the frequencies of these lineages differed between Omanis and foreigners." 
The use of MIRU-VNTR on some of the shared spoligotype clusters identify the clustered isolates as distinct and thus not having any evidence of recent transmission.
This is however not well-conveyed through the wordings currently used. 
In particular, line 29 states this may reflect transmission, while line 30-31 states no evidence of no-going transmission was found.
In addition, the amendments by the authors still states that there is spatial heterogeneity with high diversity and yet shared sub-lineages. This is confusing as if the lineages are diverse and heterogeneous, how is it shared?

Reponses:  We agree the summary given in the abstract can be confusing, we have no clarified the major findings to that showed high level of diversity of lineages, and presence of MTB isolstes with shared spoligoprofiles (clusters) belonging to Omanis and foreigners. However, the shared spoligoprofiles were broken down into distinct MIRU-VNTR profiles, excluding ongoing transmission.  We have edited the abstract to more accurately clarify the above points.

The newly provided Figure 1 has signficantly improved the understanding of the samples collection, and procedures performed in this study.
From Figure 1, it is evident that about half (47% Omanis, and 56% foreigners) of all TB cases were successfully genotyped.  Is there a particular reason why there is about 10% difference in genotyping success rate between Omanis and foreigners? 

Reponses: This is an interesting observation, however, there were no differences in the methods of collection, processing and preparation of DNA samples obtained Omanis and foreigners. One protocol has been applied to the processing of samples form the two groups. We cannot think of any technical limitation or sampling biases related to this.  

Also, is there is a difference in demographic characteristics of those who were genoytped versus those not genotyped that could confound the clustering results? 

Response: The samples subjected to spoligityping were randomly selected from the total cases included in the study, there were no province/year/nationality preferences.  

The information that spoligotyping was carried out on only successfully cultured MTB isolates, and MIRU-VNTR genotyping was performed on randomly selected samples in the major spoligotype clusters should be explictly stated in the Methods section.

Response:  Material and Methods, Section 2.2 “Spoligotyping and MIRU-VNTR typing” has been edited and extra information were added to describe selection criteria for MTB isolates that have been subjected to Spoligotyping and MIRU-VNTR analysis (Line 104 to 107 and line114 to 118).

Table 3: P-value for Sub-lineages is not possible to be 0.000.
Please correct.

Response: We thank the reviewer this has been corrected Minor Comments
--------------
Line 61: "Between 2010 to 2016,, ..."
Remove extra comma

Line 197: "CI, 01.20-2.96: should be "1.20-2.96". 

Line 243: vatiable -> variabl

Response: We thank the reviewer for noting the above typos, all have been corrected in the revised version

Reviewer 2 Report

Line 15: The abbreviation «TB» has not been introduced.

Line 17, also line 59/60: 10^5

Line 19, also line 22/28/32: Mycobacterium tuberculosis -> italics

Line 23: 4824 spoligotypes? This is more than the total number of cases analysed in this study.

Lines 23-25: Please change the wording of the sentence «A total of 482…», the meaning is not clear.

Line 24 mentions 943 clusterd isolates, figure 1 counts 927 clusterd isolates. Please check numbers.

Line: 26: The newly introduced sentence «However, the frequencies…» does not make sense and contradicts the following sentence line 28 «Diverse M. tuberculsosis lineages…»

Line 95: A total number of 2539 MTB isolates was analysed. The manuscript suggests that this is the total number of MTB isolates for the years 2009 to 2018. According to the response to the reviewes’ comments this is only the number of MTB isolates with DST. This information has to be included into the manuscript. In addition, please give the total number of MTB isolates including M. bovis and isolates without DST. For the calculateion of the annual incidences the total number of TB cases has to be used. If not all MTB cases are inclued into the study there is a strong bias on the incidences.

Line 97: Thanks for the new figure 1. But nevertheless the authors should explain in detail in the material and methods section (e.g. lines 104-117) how the selection of the strains for spoligo and MIRU typing was done. Only 1295 of 2539 strains were subjected to spoligotyping, and only for 192 of 943 clusterd strains MIRU typing was done. Why? The selection process has tob e described in the material and methods. A possible selection bias has to be discussed.

Lines 188-190: The number of strain of minor lineages is not consistent in the text and table 1. E.g. Cameroon n=9 in the text, but n=7 in table 1. Check numbers again.

Author Response

Reviewer 2.

Line 15: The abbreviation «TB» has not been introduced.

Response: This has been corrected

Line 17, also line 59/60: 10^5

Response: As detailed in the response to reviewer 1, we have changes the value “105”  to “100,000” throughout the text

Line 19, also line 22/28/32: Mycobacterium tuberculosis -> italics

Reponses: This has been edited

Line 23: 4824 spoligotypes? This is more than the total number of cases analysed in this study.

Response: We thank the reviewer for noting this, “4824” should read  “484”

Lines 23-25: Please change the wording of the sentence «A total of 482…», the meaning is not clear.

Response: We thank the reviewer, this sentence and other parts of the abstract were edited to put the results into a clearer, and more accurate form.

Line 24 mentions 943 clusterd isolates, figure 1 counts 927 clusterd isolates. Please check numbers.

Response: The information in Fig 1 has been corrected, and edited. 

Line: 26: The newly introduced sentence «However, the frequencies…» does not make sense and contradicts the following sentence line 28 «Diverse M. tuberculsosis lineages…»

Response: As detailed in the response to reviewer 1, we have edited this part of the abstract to more accurately describe our findings.  

Line 95: A total number of 2539 MTB isolates was analysed. The manuscript suggests that this is the total number of MTB isolates for the years 2009 to 2018. According to the response to the reviewes’ comments this is only the number of MTB isolates with DST. This information has to be included into the manuscript. In addition, please give the total number of MTB isolates including M. bovis and isolates without DST. For the calculateion of the annual incidences the total number of TB cases has to be used. If not all MTB cases are inclued into the study there is a strong bias on the incidences.

Response:  Information on total number of cases examined and TB incidence were revised, more details on the number of isolates excluded from the study are given in lines 101-103, and the TB incidence rates were edited (lines 182-190).

Line 97: Thanks for the new figure 1. But nevertheless the authors should explain in detail in the material and methods section (e.g. lines 104-117) how the selection of the strains for spoligo and MIRU typing was done. Only 1295 of 2539 strains were subjected to spoligotyping, and only for 192 of 943 clusterd strains MIRU typing was done. Why? The selection process has tob e described in the material and methods. A possible selection bias has to be discussed.

Response: As detailed in the response to reviewer 1, Material and Methods, Section 2.2 “Spoligotyping and MIRU-VNTR typing” has been edited and extra information were added to describe selection criteria for MTB isolates that have been subjected to Spoligotyping and MIRU-VNTR analysis.

Lines 188-190: The number of strain of minor lineages is not consistent in the text and table 1. E.g. Cameroon n=9 in the text, but n=7 in table 1. Check numbers again.

Response: We thanks the reviewer, we have checked and corrected all numbers of major and minor lineages reported in the text and Table 1.